# Make One-Shot Video Object Segmentation Efficient Again

**Tim Meinhardt**
Technical University of Munich
`tim.meinhardt@tum.de`

**Laura Leal-Taixé**
Technical University of Munich
`leal.taixe@tum.de`

## Abstract

Video object segmentation (VOS) describes the task of segmenting a set of objects in each frame of a video. In the semi-supervised setting, the first mask of each object is provided at test time. Following the one-shot principle, fine-tuning VOS methods train a segmentation model separately on each given object mask. However, recently the VOS community has deemed such a test time optimization and its impact on the test runtime as unfeasible. To mitigate the inefficiencies of previous fine-tuning approaches, we present *efficient One-Shot Video Object Segmentation* (e-OSVOS). In contrast to most VOS approaches, e-OSVOS decouples the object detection task and predicts only local segmentation masks by applying a modified version of Mask R-CNN. The one-shot test runtime and performance are optimized without a laborious and handcrafted hyperparameter search. To this end, we meta learn the model initialization and learning rates for the test time optimization. To achieve an optimal learning behavior, we predict individual learning rates at a neuron level. Furthermore, we apply an online adaptation to address the common performance degradation throughout a sequence by continuously fine-tuning the model on previous mask predictions supported by a frame-to-frame bounding box propagation. e-OSVOS provides state-of-the-art results on DAVIS 2016, DAVIS 2017 and YouTube-VOS for one-shot fine-tuning methods while reducing the test runtime substantially.
Code is available at `https://github.com/dvl-tum/e-osvos`.

## 1 Introduction

Video object segmentation (VOS) describes a two class (foreground-background) pixel-level classification task on each frame of a given video sequence. Multiple objects are discriminated by predicting individual foreground-background pixel masks. In this work, we address a variant of VOS which is semi-supervised at test time. To this end, the ground truth foreground-background segmentation mask of the first frame is provided for each object. Machine learning methods that tackle semi-supervised VOS are categorized by their utilization of the provided object ground truth masks.

We focus on fine-tuning methods [6, 22, 39, 20, 34], which exploit the transfer learning capabilities of neural networks and follow a multi-step training procedure: (i) *pre-training steps*: learn general image and segmentation features from training the model on images and video sequences , and (ii) *fine-tuning*: one-shot test time optimization which enables the model to learn foreground-background characteristics specific to each object and video sequence. While elegant through their simplicity, fine-tuning methods face important shortcomings: (i) pre-training is fixed and not optimized for the subsequent fine-tuning, (ii) the hyperparameters of the test time optimization are often excessively handcrafted and fail to generalize between datasets. The common existing fine-tuning setups [6, 20] are inefficient and suffer from high test runtimes with as many as 1000 training iterations per segmented object. As a consequence, recent methods refrain from such an optimization at test time and

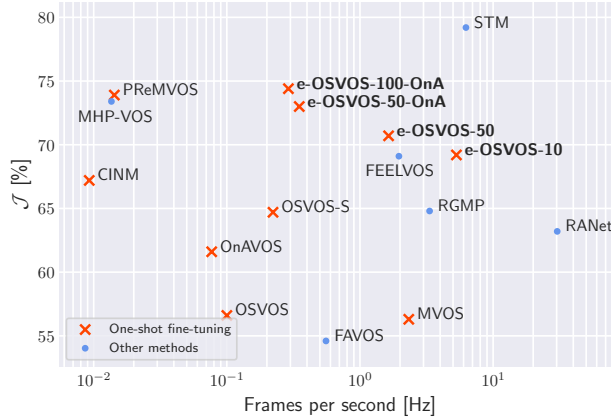

Figure 1: Performance versus runtime comparison of modern video object segmentation (VOS) approaches on the DAVIS 2017 validation set. We only show methods with publicly available runtime information. Our e-OSVOS approach demonstrates the relevance of fine-tuning for VOS and its inherent flexibility as we apply the same meta learned optimization for a varying number of iterations and with online adaptation (OnA).

instead opt for solutions such as template matching [7, 13] and mask propagation [8, 9, 24, 25, 35, 44] for semi-supervised VOS.

In this work, we revisit the concept of one-shot fine-tuning for VOS, and show how to leverage the power of meta learning to overcome the aforementioned issues. To this end, we propose three key design choices which make one-shot fine-tuning for VOS efficient again:

**Learning the Model Initialization** The common pre-training [6, 22, 20, 34] yields a segmentation model not specifically optimized for the subsequent fine-tuning task and requires an *unlearning* of potential false positive objects. Therefore, we propose to meta learn the pre-training step, i.e., we learn the best initialization of the segmentation model for a subsequent fine-tuning to any object.

**Learning Neuron-Level Learning Rates** We replace the laborious and handcrafted hyperparameter search from [6, 22, 20, 34] and additionally optimize learning rates for each neuron of the model. In contrast to a single learning rate for the entire model [1] or millions for all of its parameters [39], this allows for an ideal balance between individual learning behavior and additional trainable parameters.

**Optimization of Model with Object Detection** To account for the foreground-background pixel imbalance and the challenging object discrimination by individual fine-tuning, previous fine-tuning methods [22, 20, 34] rely on additional mask proposal or bounding box prediction methods. In contrast, we directly fine-tune Mask R-CNN [11] with its separate end-to-end trainable object detection head which limits mask predictions to local object bounding boxes.

This leads to our *efficient one-shot video object segmentation* (e-OSVOS) approach, which achieves state-of-the-art segmentation performance on the DAVIS-2016, DAVIS-2017 and YouTube-VOS benchmarks compared to all previous fine-tuning methods, at a much lower test runtime, see Figure 1. Overall, our results combat the negative preconceptions with respect to fine-tuning as a principle for semi-supervised VOS, and are intended to motivate future research in this direction.

## 1.1 Related Work

We categorize VOS methods by their application of one-shot fine-tuning for semi-supervised VOS.

**Without Fine-Tuning** Several methods [7, 13, 33] pose VOS as the task of pixel retrieval in the learned embedding space. After the embedding learning, no fine-tuning is necessary during the inference – pixels are simply their respective nearest neighbors in the learned embedding space [7, 13] or used as a guide to the segmentation network [33]. Other methods propagate segmentation masks using optical flow or point trajectories [9, 35, 36] or segment, propagate and combine object parts [8]. The authors of [24] propagate and decode segmentation masks based on the first- and query-frame

embeddings. STM [31] leverages a memory network to capture the information of the object in the past frames which is then decoded to predict the current frame mask. They achieve state-of-the-art performance but fail to capture small objects and require a large GPU memory for sequences with many objects.

**With Fine-Tuning** The concept of fine-tuning for semi-supervised VOS was first introduced in OSVOS [6]. This family of methods fine-tunes a pre-trained segmentation model to the first frame ground truth mask of a given object and predicts segmentation masks for the remaining video frames. OnAVOS [34] extends this approach by adapting the target appearance model online on consecutive frames using heuristics-based fine-tuning policies. While conceptually elegant, the aforementioned methods have no notion of individual objects, shapes, or motion consistency. To remedy this issue, OSVOS-S [22] and PReMVOS [21] leverage object detection and instance segmentation methods (*e.g.*, MaskRCNN [11]) during the inference as additional object guidance cues. This approach is akin to the *tracking-by-detection* paradigm, commonly followed in the multi-object tracking community. Fine-tuning methods for VOS all share one major drawback – the online fine-tuning process requires extensive manual hyperparameter search and so far numerous training iterations during the inference (up to 1000 in the original OSVOS [6] method). Hence, recent methods refrain from such an optimization at test time due to its impact on the runtime.

**Towards Efficient Fine-Tuning** Ideally, we would like to learn an appearance model and perform as-few-as-possible training steps to during the inference. One viable approach consists of posing the video object segmentation task as a meta learning problem and optimizing the fine-tuning policies (*e.g.*, generic model initialization, learning rates, and the number of fine-tuning iterations). The first attempt in this direction, MVOS [39], proposes to learn the initialization and learning rates per model parameter. However, this approach is impractical for modern large-scale detection/segmentation networks. In this paper, we revisit the concept of meta learning for VOS and propose several critical design choices which yield state-of-the-art results and vastly outperform MVOS [39] and other fine-tuning methods.

**Meta Learning for Few-Shot Learning.** Previous works have addressed analogous issues for image classification. The authors of MAML [10] propose to learn the model initialization for an optimal subsequent fine-tuning at test time. Such an initialization is supposed to benefit the fine-tuning beyond the traditional transfer learning which merely internalizes the training data. The MAML++ [1] and Meta-SGD [16] approaches suggest several improvements to MAML and compliment the model initialization by learning the optimal learning rate. However, both approaches limit their potential by optimizing only a single global learning rate for the entire model. The authors of [45] conduct an analysis of the meta learning for few-shot scenarios problem and address the memorization problem with a specifically tailored loss function. Other approaches, such as [29], suggest to not only predict the learning rate but apply a parameterized model to predict the entire update step. However, these approaches so far are limited in their applicability to large scale neural networks.

## 2 One-Shot Fine-Tuning for Video Object Segmentation

For a given video sequence $n$ with $I_n$ image frames $\{\mathbf{x}_i^n : 0 \leq i < I_n\}$ and $K_n$ objects, video object segmentation (VOS) predicts individual object masks $\mathbf{y}_i^{n,k}$ of all frames $\mathbf{x}_i^n$. In the case of semi-supervised VOS, the ground truth mask $\hat{\mathbf{y}}_0^{n,k}$ of a single frame is provided at test time for each object. For simplicity, we assume that the given frame always corresponds to the first frame $i = 0$ of the video. However, potentially a video might contain multiple objects entering the sequence at different frames. The common approach for one-shot fine-tuning of a segmentation model $f$ and its parameters $\boldsymbol{\theta}_f$ follows the three step optimization pipeline presented in [6]: (i) *Base network:* Learn general object features by training the feature extractor backbone of $f$ on a large scale image recognition challenge, e.g. ImageNet [30]. (ii) *Parent network:* Train $f$ on a segmentation dataset, e.g., DAVIS-17 training set [28], to learn the foreground-background segmentation problem. (iii) *Fine-tuning:* Learn object and sequence specific features by separately fine-tuning the parent network to each object of a given video sequence. It should be noted that one-shot learning by nature runs full-batch updates, hence iteration and epoch are often used interchangeably. For a sequence $n$, the fine-tuning yields $K_n$ separately trained models $f^{n,k}$ with parameters $\boldsymbol{\theta}_f^{n,k}$. The final object masks $\mathbf{y}_i^{n,k} = f^{n,k}(\mathbf{x}_i^n)$ are obtained from the maximum over the predicted pixel probabilities over all objects $K_n$. The steps (ii) and (iii) minimize the segmentation loss $\mathcal{L}_{seg}(\mathcal{D}, \boldsymbol{\theta}_f)$, e.g., binary

cross-entropy, of the model $f$ on a given training dataset $\mathcal{D}$. For clarity, we omit the sequence and object indices on $f$ and $\boldsymbol{\theta}_f$ in future references and refer to a problem solved by an optimization $g$ as:

$$\boldsymbol{\theta}_f^g = \underset{\boldsymbol{\theta}_f \text{ with } g}{\operatorname{argmin}} \mathcal{L}_{seg}(\mathcal{D}, \boldsymbol{\theta}_f) \tag{1}$$

Such an optimization is defined by several hyperparameters, including, the model initialization, number of training iterations and the type of stochastic gradient descent (SGD) method as well as its learning rate(s).

# 3 Efficient One-Shot Video Object Segmentation

We describe the key design choices of e-OSVOS, namely, the model choice, meta learning of the fine-tuning optimization, and two additional test-time modifications to further enhance our performance.

## 3.1 Optimization of Model with Object Detection

Fine-tuning a fully-convolutional model for VOS suffers from two major issues: (i) the imbalance between foreground and background pixels and (ii) the challenging object discrimination by individual fine-tuning. Typically, the latter requires an *unlearning* of potential false positive pixels. Several fine-tuning approaches [22, 20, 34] tackle these issues by including separate mask proposal or bounding box prediction methods. We propose to directly fine-tune Mask R-CNN [11] which decouples the object detection and requires the demanding pixel-wise segmentation only to bounding boxes.

Mask R-CNN consists of a feature extraction backbone, a Region Proposal Network (RPN) and two network heads, namely, the bounding box object detection and mask segmentation heads. The RPN produces potential bounding box candidates, also known as proposals, on the intermediate feature representation provided by the backbone. The box detection head predicts the object class and regresses the final bounding boxes for each proposal. Finally, the segmentation head provides object masks for each object class and bounding box. The segmentation loss mentioned in Section 2 corresponds to the multi-task Mask R-CNN loss: $\mathcal{L}_{seg} = \mathcal{L}_{RPN} + \mathcal{L}_{box} + \mathcal{L}_{mask}$.

We adapt Mask R-CNN for the VOS task by replacing the pixel-wise cross-entropy $\mathcal{L}_{mask}$ loss with the Lovász-Softmax [5] loss. The Lovász-Softmax loss directly optimizes the intersection-over-union and demonstrates superior performance in our one-shot fine-tuning setting. In contrast to commonly applied batch normalization, group normalization [38] allows for a fine-tuning even on single sample (frame) batches. Therefore, we replace all normalization layers of the backbone with group normalization.

## 3.2 Meta Learning the One-Shot Test Time Optimization

As outlined by [29], meta learning is of particular interest for semi-supervised or few-shot learning scenarios. In this work, we extend this idea from image classification to VOS and meta learn step (ii) and (iii) of the optimization pipeline from Section 2. To this end, we learn differentiable components of the test time optimization, specifically, the model initialization and SGD learning rate(s).

### 3.2.1 Meta Tasks

In order to meta learn the optimization, we formulate the VOS fine-tuning problem as a meta task. A task represents the fine-tuning optimization on a single object of a video sequence. Given a set of $N$ unique video sequences each with $K_n$ objects, we define the corresponding taskset $\mathcal{T} = \{T_{n,k} : 0 \leq k < K_n | 0 \leq n < N\}$ with $T_{n,k} = \{\mathcal{D}_{train}^{n,k}, \mathcal{D}_{test}^{n,k}\}$. As illustrated in Figure 2a, an individual task is created by splitting each sequence into a training and test dataset consisting of disjoint sets of video frames. The goal of task $T_{n,k}$ is to minimize the test loss $\mathcal{L}_{seg}(\mathcal{D}_{test}^{n,k}, f)$ of the model $f$. The datasets $\mathcal{D}_{train}^{n,k} = \{\mathbf{x}_0^n, \hat{\mathbf{y}}_0^{n,k}\}$ and $\mathcal{D}_{test}^{n,k} = \{\{\mathbf{x}_i^n, \hat{\mathbf{y}}_i^{n,k}\} : 1 \leq i < I_n\}$ include the first and all consecutive frames, respectively. We train e-OSVOS on the taskset $\mathcal{T}_{train}$ such that the fine-tuning optimization on any $\mathcal{D}_{train}^{n,k}$ yields optimal results on the corresponding $\mathcal{D}_{test}^{n,k}$. This involves two optimizations, namely, the inner fine-tuning and outer meta optimization. As for all machine learning methods, the final generalization to the test taskset $\mathcal{T}_{test}$ is paramount. In future references of the datasets $\mathcal{D}_{train}$ and $\mathcal{D}_{test}$ we again omit the sequence and object indices $n, k$.

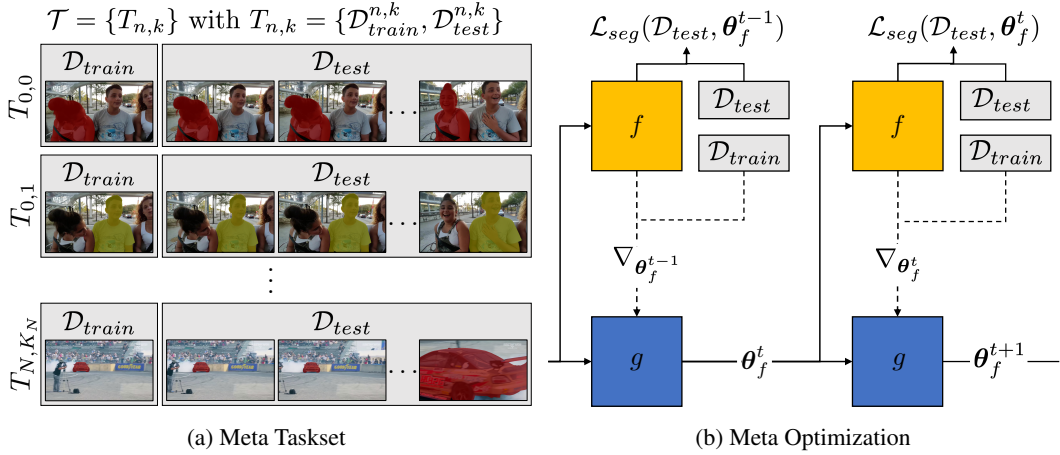

| (a) Meta Taskset | (b) Meta Optimization |

Figure 2: The test time optimization $g$ of e-OSVOS is meta learned on a VOS taskset structured as in (a). Each task represents a video sequence with its frames split into training $\mathcal{D}_{train}^n$ and test $\mathcal{D}_{test}^n$ datasets. The optimization $g$ depicted in (b) consists of the model initialization and a set of learning rates applied with vanilla stochastic gradient descent. Both of which are meta learned by backpropagating the final test loss $\mathcal{L}_{seg}(\mathcal{D}_{test}, \boldsymbol{\theta}_f^T)$.

### 3.2.2 Meta Optimization

In analogy to *How to train your MAML* [1], our test time optimization consists of a vanilla SGD with two trainable components, namely, the initialization of the model $f$ and learning rates $\boldsymbol{\lambda}$ which are applied for a fixed number of iterations. We refer to the trainable parameters of such an optimization $g$ with $\boldsymbol{\theta}_g$. Learning a single task involves the following bi-level optimization problem for $\boldsymbol{\theta}_g$ and $\boldsymbol{\theta}_f$:

$$\boldsymbol{\theta}_g^* = \underset{\boldsymbol{\theta}_g}{\arg\min}\, \mathcal{L}_{seg}(\mathcal{D}_{test}, \boldsymbol{\theta}_f^g), \tag{2}$$

$$\text{s.t. } \boldsymbol{\theta}_f^g = \underset{\boldsymbol{\theta}_f \text{ with } g}{\arg\min}\, \mathcal{L}_{seg}(\mathcal{D}_{train}, \boldsymbol{\theta}_f). \tag{3}$$

The outer optimization in Equation (2) is handcrafted and performed on batches of tasks from $\mathcal{T}_{train}$. The bi-level optimization aims to maximize the generalization from a given training to test dataset. In practice, one step of Equation (2) includes multiple steps in Equation (3). This corresponds to fine-tuning the model $f$ for multiple iterations on the first frame $\mathcal{D}_{train}$. The optimization $g$ is trained by *Backpropagation Through Time* (BPTT) of the test loss after $T$ training iterations:

$$\mathcal{L}_{BPTT} = \mathcal{L}_{seg}(\mathcal{D}_{test}, \boldsymbol{\theta}_f^T), \tag{4}$$

$$\text{with } \boldsymbol{\theta}_f^{t+1} = g(\nabla_{\boldsymbol{\theta}_f^t} \mathcal{L}_{seg}(\mathcal{D}_{train}, \boldsymbol{\theta}_f^t), \boldsymbol{\theta}_f^t). \tag{5}$$

As illustrated in Figure 2b, $g$ connects the computational graph of each iteration over time. The optimization applies a gradient descent step with respect to $\mathcal{D}_{train}$ and updates $f$. To this end, the optimization receives the current model parameters $\boldsymbol{\theta}_f^t$ and their gradients $\nabla_{\boldsymbol{\theta}_f^t} \mathcal{L}_{seg}(\mathcal{D}_{train}, \boldsymbol{\theta}_f^t)$. After $T$ updates, the optimization itself is updated to minimize $\mathcal{L}_{BPTT}$ with respect to the updated model $f$. As Equation (5) already requires the computation of model parameter gradients, the outer backpropagation of $\mathcal{L}_{BPTT}$ introduces second order derivatives. To reduce the computational effort, these can be omitted which is equivalent to ignoring the dashed edges of the graph in Figure 2b.

### 3.2.3 Learning the Segmentation Model Initialization

Meta learning the model initialization for a subsequent optimization (fine-tuning) yields superior performance compared to classic transfer learning approaches (parent network training). The initialization not only internalizes the data of the tasks, but benefits the subsequent fine-tuning step. Previous works [10, 16, 39], have applied this successfully to few-shot image classification. For semi-supervised VOS, meta learning is able to provide a model initialization $\boldsymbol{\theta}_f^0$ for the fine-tuning

optimization in Equation ([5](#)). Such an initialization avoids biases for specific objects and eases the individual fine-tuning to each object significantly. In addition to the classic overfitting to $\mathcal{T}_{train}$, meta learning is prone to zero-shot collapsing, also called the *memorization problem* [45]. For image classification, this is avoided by randomly shuffling the class labels for each training task. For multi-object VOS we tackle the issue of zero-shot collapsing by separating objects of the same sequence to multiple tasks. The first two example tasks in Figure [2a](#) demonstrate the necessity of an one-shot optimization for segmenting different objects given the same input image.

### 3.2.4 Learning Neuron-Level Learning Rates

The optimization $g$ performs Equation ([5](#)) with a vanilla SGD step and updates the segmentation model by applying a set of meta learned learning rates $\boldsymbol{\lambda}$ for a fixed number of iterations. The entire set of trainable optimization parameters is denoted as $\boldsymbol{\theta}_g = \{\boldsymbol{\theta}_f^0, \boldsymbol{\lambda}\}$. Previous meta learning for few-shot approaches applied learning rates at varying parameter hierarchy levels, from a single global learning rate for the entire model in [1], to learning rates for all model parameters $\boldsymbol{\theta}_f$ in MVOS [39]. The latter is unfeasible for many modern state-of-the-art segmentation networks as it effectively doubles the number of trainable optimization parameters ($|\boldsymbol{\theta}_g| \approx 2|\boldsymbol{\theta}_f^0|$).

Therefore, we propose an ideal balance between individual learning behavior and additional trainable parameters by optimizing a set of learning rates at the neuron level. A common linear neural network layer consists of multiple neurons, or kernels for convolution layers, where each neuron applies a weight tensor and corresponding scalar bias. We predict a pair of learning rates for each neuron of the model $f$, i.e., a single rate for each weight tensor and bias scalar. The amount of additional trainable parameters is neglectable for modern segmentation models as their total number of parameters typically exceeds $10^7$. In Algorithm 1 of the supplementary, we illustrate the full e-OSVOS training pipeline for a given VOS taskset $\mathcal{T}_{train}$.

## 3.3 Online Adaption and Bounding Box Propagation

By nature, fine-tuning methods are prone to overfit on the given single frame dataset $\mathcal{D}_{train}^{n,k} = \{\mathbf{x}_0^n, \hat{\mathbf{y}}_0^{n,k}\}$. For sequences with changing object appearance or new similar objects entering the scene, such an overfitting often results in degrading recognition performance or drifting of the segmentation mask. However, e-OSVOS incorporates two test time techniques to overcome those problems.

**Online adaptation** Inspired by [34], we apply an online adaptation (OnA) which continuously fine-tunes the segmentation model on both the given first frame ground truth and past mask predictions. First, we fine-tune the model for $T$ iterations only on the first frame which yields $\boldsymbol{\theta}_f^T$ and then continue the fine-tuning every $I_{OnA}$ frames for $T_{OnA}$ additional iterations on the combined online dataset $\mathcal{D}_{train}^{n,k} = \{\mathbf{x}_0^n, \hat{\mathbf{y}}_0^{n,k}\} \cup \{\mathbf{x}_i^n, \mathbf{y}_i^{n,k}\}$. In contrast to [34], our efficient test time optimization allows for a reset of the model before every additional fine-tuning to the first-frame model state $\boldsymbol{\theta}_f^T$. Such a reset avoids the accumulation of false positive pixels wrongly considered as ground truth. Our learned optimization $g$ generalizes to such an online adaptation without any additional meta learning.

**Bounding Box Propagation** In analogy to [4], we extend the RPN proposals with the detected object boxes of the previous frame. To account for the changing position of the object, we augment the previous boxes with random spatial transformations. Starting with the first frame ground truth boxes, the frame-to-frame propagation facilitates the tracking of each object over the sequence.

## 4 Experiments

We demonstrate the applicability of e-OSVOS on three semi-supervised VOS benchmarks, namely, DAVIS 2016 [27], DAVIS 2017 [28] and YouTube-VOS [41]. The tasksets $\mathcal{T}$ for training and evaluation of e-OSVOS are constructed from the corresponding training, validation and test video sequences of each benchmark.

### 4.1 Datasets and Evaluation Metrics

**DAVIS 2016** The DAVIS 2016 [27] benchmark consists of a training and validation set with 30 and 20 single object video sequences, respectively. Every sequence is captured at 24 frames per second (FPS) and semi-supervision is achieved by providing the respective first frame object mask.

**DAVIS 2017** The DAVIS 2017 [28] benchmark extends DAVIS-16 with 100 additional sequences including dedicated test-dev and test sets. The validation, test-dev and test sets each consist of 30 sequences. The extended train set contains the remaining 60 video sequences. In addition, DAVIS 2017 contains a mix of single and multi-object sequences with varying image resolutions.

**YouTube-VOS** Our largest benchmark, YouTube-VOS [41], consists of 4453 video sequences including dedicated test and validation sets with 508 and 474 sequences, respectively. As DAVIS 2017, this benchmark contains single and multi-object sequences in multiple resolutions but provides segmentation ground truth only at 6 FPS. In general, [41] requires stronger tracking capabilities as objects enter in the middle of the sequence or leave and reenter the frame entirely.

**Evaluation Metrics** We evaluate the standard VOS metrics defined by [27]. For the Intersection over Union between predicted and ground truth masks, also known as Jaccard index $\mathcal{J}$ in %, we evaluate the mean as well as decay over all frames. Furthermore, we report the mean contour accuracy $\mathcal{F}$ in %, the combination of both mean metrics $\mathcal{J}\&\mathcal{F}$ in % and the frames per second (FPS) in Hz.

### 4.2 Implementation Details

We conduct all experiments on a Mask R-CNN model with ResNet50 [12] and FPN [17] pre-trained on the COCO [18] segmentation dataset. In order to optimize the learning rates and model initialization jointly without overfitting, we follow previous VOS approaches such as [33, 23] and train e-OSVOS on YouTube-VOS combined with DAVIS 2017. To improve generalization, we construct training tasks $T_{n,k} \in \mathcal{T}_{train}$ by randomly sampling a single frame from a sequence and augmenting it with spatial and color transformations for each train and test dataset. Furthermore, for both DAVIS datasets, we fine-tune the meta learning of the model initialization for each dataset while keeping the previously learned learning rates fixed. For the outer optimization we apply RAdam [19] with a fixed learning rate $\beta$, as shown in Algorithm 1 of the supplementary, on batches of 4 training tasks each distributed to a Quadro RTX 6000 GPU for a total of 4 days. To limit the computational effort, we ignore second order derivatives and fine-tune for $T = 5$ BPTT iterations. The learning rates are clamped to be non-negative after each meta update.

The online adaptation (OnA) is applied every $I_{OnA} = 5$ steps for $T_{OnA} = 10$ iterations. To further boost inner-sequence generalization, we apply spatial random transformations as in [6] during the initial fine-tuning but not for the online adaptation. While the iterations are fixed to $T = 5$ during the meta learning e-OSVOS generalizes to varying numbers of iterations and the online adaptation without any further learning. To indicate different versions of e-OSVOS, we denote the number of initial fine-tuning iterations and if online adaptation is applied.

### 4.3 Ablation Study

We demonstrate the effect of the individual e-OSVOS components on the DAVIS 2017 validation set in Table 1. Both the parent and meta training utilize the combined dataset of YouTube-VOS and DAVIS 2017. For a fair comparison between varying number of fine-tuning iterations, we refrained from any spatial random transformations at test time. The first row shows a handcrafted equivalent of the e-OSVOS test time optimization for which we apply a grid search to find the optimal global fine-tuning learning rate. Note, this baseline is not representative for state-of-the-art fine-tuning VOS approaches as we omitted any additional handcrafted test time improvements [6, 22, 20, 34], such as, layer-wise learning rates, learning rate scheduling, contour snapping. The handcrafted approach is inferior to meta learning the initialization and a single global learning rate even for substantially more iterations. The neuron-level learning rates and additional modifications to the Mask R-CNN motivated in Section 3 both yield substantial segmentation performance gains. While the improvement from bounding box propagation is comparatively small, it only adds an insignificant amount of additional runtime. The marginal improvement from e-OSVOS-50 to e-OSVOS-100 clearly motivates the application of an online adaption to combat overfitting to the first frame and degrading performance over the course of the sequence. In Figure 3, we further demonstrate the efficiency of e-OSVOS on

Table 1: **Ablation study** of each e-OSVOS component on the DAVIS 2017 validation set. The first row represent a handcrafted equivalent of our test time optimization. We present performance gains componentwise for 10 fine-tuning iterations and iterationwise for our final e-OSVOS version.

| Method | Iterations ($T$) | $\mathcal{J}\&\mathcal{F}\uparrow$ | |
|---|---|---|---|
| Mask R-CNN | 10 | 33.6 | |
| | 50 | 39.7 | |
| + parent training + single LR search | 100 | 41.6 | |
| | 1000 | 42.7 | |
| Mask R-CNN | | | |
| + Learn model initialization and single LR | 10 | 64.4 | + 30.8 |
| + Learn neuron level learning rates | 10 | 67.2 | + 2.8 |
| + Group normalization + Lovász-Softmax | 10 | 69.4 | + 2.2 |
| + Bounding box propagation (e-OSVOS) | 10 | 69.9 | + 0.5 |
| + Online adaption (e-OSVOS-OnA) | 10 | 71.2 | + 1.3 |
| e-OSVOS-$T$ | 50 | 71.3 | |
| | 100 | 71.2 | |
| e-OSVOS-$T$-OnA | 50 | 73.7 | |
| | 100 | 74.8 | |

Figure 3: We evaluate e-OSVOS for increasing number of initial fine-tuning iterations $T$ on the DAVIS 2017 validation set. The first iterations yield the largest performance gains while still running at comparatively large frames per second rates.

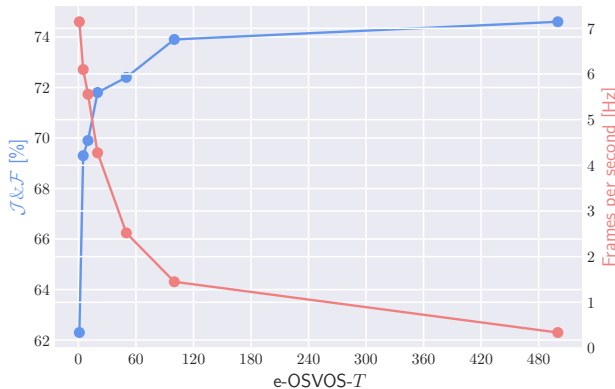

the DAVIS 2017 validation set. The meta learning enables large gains in segmentation performance after only a few fine-tuning iterations without suffering from low frames per second rates. With increasing number of iterations the actual inference time of the sequence becomes neglectable.

## 4.4 Benchmark Evaluation

We present state-of-the-art VOS results for fine-tuning methods on DAVIS 2016 and 2017 in Table 2 and for YouTube-VOS in Table 3. We focus our evaluation on fine-tuning, hence separating the results of methods without fine-tuning (FT). The overall state-of-the art method STM [31], which does not leverage fine-tuning, currently surpasses all existing approaches in terms of performance and runtime. Nevertheless, we want to motivate fine-tuning as a concept which is applicable to further boost results of methods like STM without harming its efficiency.

**DAVIS 2016 and 2017** In terms of the important $\mathcal{J}$ metric, we outperform all previous one-shot fine-tuning approaches on the validation set while reducing the runtime multiple orders of magnitude. It is important to note, that unlike our approach all previous fine-tuning methods rely on post processing or an ensemble of models to achieve optimal results. We even surpass PReMVOS [21], the long-time state-of-the-art VOS method, with a much simpler and more efficient fine-tuning approach. PReMVOS applies an additional contour snapping, as in [6], which explains its superiority in terms of contour accuracy $\mathcal{F}$. On the test-dev set all methods achieve substantially worse results in all metrics compared to the validation set. This is due to sequences more challenging with respect to instance identity preservation. We do not achieve state-of-the-art results for fine-tuning methods on

Table 2: VOS performance evaluation on the **DAVIS 2016 and 2017** benchmarks. We categorize methods by their application of fine-tuning (FT) and post-processing (PP) of the predicted masks and label methods with an ensemble of models with †. The table is ordered by $\mathcal{J}\&\mathcal{F}$ on DAVIS 2017 validation. The evaluation metrics are detailed in Sec. 4.1. If not publicly available we adopted runtimes (FPS) from [3].

| Method | FT | PP | DAVIS 2016 - validation $\mathcal{J}\uparrow$ | $\mathcal{J}$ Decay$\downarrow$ | $\mathcal{F}\uparrow$ | FPS$\uparrow$ | DAVIS 2017 - validation $\mathcal{J}\uparrow$ | $\mathcal{J}$ Decay$\downarrow$ | $\mathcal{F}\uparrow$ | $\mathcal{J}\&\mathcal{F}\uparrow$ | DAVIS 2017 - test-dev $\mathcal{J}\uparrow$ | $\mathcal{J}$ Decay$\downarrow$ | $\mathcal{F}\uparrow$ | $\mathcal{J}\&\mathcal{F}\uparrow$ |
|---|---|---|---|---|---|---|---|---|---|---|---|---|---|---|
| FAVOS [8] | | × | 82.4 | **4.5** | 79.5 | 0.56 | 54.6 | 14.1 | 61.8 | 58.2 | 42.9 | 18.1 | 44.2 | 43.6 |
| RGMP [24] | | | 81.5 | 10.9 | 82.0 | **7.70** | 64.8 | 18.9 | 68.6 | 66.7 | 51.3 | 34.3 | 54.4 | 52.8 |
| RVOS [24] | | | – | – | – | – | 57.5 | 24.9 | 63.6 | 60.6 | 47.9 | 35.7 | 52.6 | 50.3 |
| MetaVOS [3] | | | 81.5 | 5.0 | 82.7 | 4.0 | 63.9 | 14.4 | 70.7 | 67.3 | – | – | – | – |
| RANet [37] | | | 86.6 | 7.4 | 87.6 | 30.3 | 63.2 | 18.6 | 68.2 | 65.7 | 53.4 | 21.9 | 57.3 | 55.4 |
| FEELVOS [33] | | | 81.1 | 13.7 | 82.2 | 2.22 | 69.1 | 17.5 | 74.0 | 71.5 | 55.1 | 29.8 | 60.4 | 57.8 |
| MHP-VOS [42] | | | 87.6 | 6.9 | 89.5 | 0.01 | 73.4 | 17.8 | 78.9 | 76.1 | 66.4 | 18.0 | 72.7 | 69.5 |
| STM [31] | | | **88.7** | 5.0 | **90.1** | 6.25 | **79.2** | **8.0** | **84.3** | **81.7** | 69.3 | 16.9 | 75.2 | 72.2 |
| CINM† [2] | × | × | 83.4 | 12.3 | 85.0 | 0.01 | 67.2 | 24.6 | 74.4 | 70.7 | 64.5 | 20.0 | 70.5 | 67.5 |
| Lucid [15] | × | × | 83.9 | 9.1 | 82.0 | 0.005 | – | – | – | – | 63.4 | 19.5 | 69.9 | 66.6 |
| MVOS [39] | × | × | 83.3 | – | 84.1 | 4.0 | 56.3 | – | 62.1 | 59.2 | – | – | – | – |
| OSVOS [6] | × | × | 79.8 | 14.9 | 80.6 | 0.11 | 56.6 | 26.1 | 63.9 | 60.3 | 47.0 | 19.2 | 54.8 | 50.9 |
| OSVOS-S† [22] | × | × | 85.6 | 5.5 | 87.5 | 0.22 | 64.7 | 15.1 | 71.3 | 68.0 | 52.9 | 24.1 | 62.1 | 57.5 |
| OnAVOS [34] | × | × | 86.1 | 5.2 | 84.9 | 0.08 | 61.6 | 27.9 | 69.1 | 65.3 | 49.9 | 23.0 | 55.7 | 52.8 |
| PReMVOS† [21] | × | | 84.9 | 8.8 | **88.6** | 0.01 | 73.9 | 16.2 | **81.8** | 77.8 | 67.5 | 21.7 | 75.8 | 71.6 |
| e-OSVOS-10 | × | | 85.1 | 5.0 | 84.8 | **5.3** | 69.2 | 18.5 | 74.6 | 71.9 | – | – | – | – |
| e-OSVOS-50 | × | | 85.5 | 5.0 | 85.8 | 1.64 | 70.7 | 18.6 | 75.9 | 73.3 | – | – | – | – |
| e-OSVOS-50-OnA | × | | 85.9 | 5.2 | 85.9 | 0.35 | 73.0 | 13.6 | 78.3 | 75.6 | 60.9 | 22.1 | 68.6 | 64.8 |
| e-OSVOS-100-OnA | × | | **86.6** | **4.5** | 87.0 | 0.29 | **74.4** | **13.0** | 80.0 | 77.2 | – | – | – | – |

Table 3: VOS performance evaluated on the **YouTube-VOS** validation set. This benchmark additionally evaluates the performance on completely unseen object classes. Results of other methods are copied from [31].

| Method | FT | PP | YouTube-VOS - validation Overall$\uparrow$ | $\mathcal{J}$ Seen$\uparrow$ | $\mathcal{F}$ Seen$\uparrow$ | $\mathcal{J}$ Unseen$\uparrow$ | $\mathcal{F}$ Unseen$\uparrow$ |
|---|---|---|---|---|---|---|---|
| OSMN [43] | | | 51.2 | 60.0 | 60.1 | 40.6 | 44.0 |
| MSK [26] | | | 53.1 | 59.9 | 59.5 | 45.0 | 47.9 |
| RGMP [24] | | | 53.8 | 59.5 | – | 45.2 | – |
| RVOS [32] | | | 56.8 | 63.6 | 67.2 | 45.5 | 51.0 |
| S2S [40] | | | 64.4 | 71.0 | 70.0 | 55.5 | 61.2 |
| A-GAME [14] | | | 66.1 | 67.8 | – | 60.8 | – |
| STM [31] | | | **79.4** | **79.7** | **84.2** | **72.8** | **80.9** |
| OnAVOS [34] | × | × | 55.2 | 60.1 | 62.7 | 46.6 | 51.4 |
| OSVOS [6] | × | × | 58.8 | 59.8 | 60.5 | 54.2 | 60.7 |
| PReMVOS† [21] | × | | 66.9 | 71.4 | **75.9** | 56.5 | 63.7 |
| e-OSVOS-50-OnA | × | | **71.4** | **71.7** | 66.0 | **74.3** | **73.8** |

the test-dev set. However, our approach still demonstrates the potential of fine-tuning as we surpass most none-fine-tuning methods without applying any post-processing or an ensemble of methods.

**YouTube-VOS** On the more challenging YouTube-VOS dataset, our approach yields overall better results compared to all previous fine-tuning methods. In particular, PReMVOS suffers from inferior performance on unseen object classes. This indicates that our meta learned initialization provides a superior fine-tuning initialization which is less prone to overfitting. It should be noted that some methods were evaluated on an earlier version of the YouTube-VOS benchmark which causes slight variations in the final results.

## 5 Conclusion

This works demonstrates the application of meta learning to VOS fine-tuning and makes one-shot video object segmentation efficient again. We first motivate our model choice to be a modified Mask R-CNN instead of a fully convolutional segmentation model. Furthermore, we meta learn the model initialization and a set of neuron-level learning rates. e-OSVOS works in addition to common test-time techniques which mitigate performance degradation, such as online adaptation with continuous fine-tuning and a bounding box propagation. We demonstrate the best performance amongst fine-tuning methods, and aspire to reignite research in this promising approach to semi-supervised VOS.

**Acknowledgements**

This research was funded by the Humboldt Foundation through the Sofja Kovalevskaja Award.

**Broader Impact**

Authors are asked to include a section in their submissions discussing the broader impact of their work, including possible societal consequences, both, positive and negative.

Many methods for video object segmentation or multiple object tracking rely on appearance models of objects. In this work, we have shown that one can rely on the simple but elegant solution of fine-tuning of a model as a way to build appearance models.

Semi-supervised video object segmentation is often used to automatize video editing, e.g., to remove one object from a video. While it is clear that more automatic methods would have a positive impact in reducing the manual work needed to perform such video edits, there is also potential to misuse such technology. One could imagine the creation of fake videos, where objects are taken out or put on the scene to create out-of-context content that might lead viewers to misinterpret the situation. Nonetheless, we believe the technology is still in early stages and far from being able to create fake content without substantial knowledge and manual work. Therefore, we believe that, for this particular task, the benefits outweigh the potential misuses of the technology.

Appearance models are also key towards tackling multi-object tracking and segmentation, important for applications such as robotics. For example, social robots are often tasked with following one specific person, hence the robot has to learn fast an on the fly the appearance of the specific person that it has to follow. This can be extended to multiple people tracking, where each model would be fine-tuned to a specific person on the scene. Segmentation of an object of interest becomes also key for robotic tasks such as grasping or any object-robot interaction. But multi-object tracking and video object segmentation also have a dark side, with applications such as illegal surveillance. We want to note, that our method does not make use of any kind of identifying characteristic of a person (if the person would be our object to follow and segment). Therefore, we believe our technology does not directly contribute nor promote these kinds of misuses.

We believe that the simple concept of fine-tuning a model to a specific object is incredibly powerful. With our work, we hope to inspire researchers to continue with that paradigm, now that we can properly train it to achieve state-of-the-art results. Looking at the impact that these tools can have for society, one can see extremely positive things such as the realization of social robots that could help the elderly in their daily chores.

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
