[Supplementary Material]

# Make One-Shot Video Object Segmentation Efficient Again
## Supplementary Material

**Tim Meinhardt**
Technical University of Munich
`tim.meinhardt@tum.de`

**Laura Leal-Taixé**
Technical University of Munich
`leal.taixe@tum.de`

## Abstract

The supplementary material complements our work with the training algorithm of our efficient One-Shot Video Object Segmentation (e-OSVOS) and additional implementation as well as training details. Furthermore, we provide a more detailed comparison to PremVOS, a state-of-the-art fine-tuning method, including selected visual results.

## 1 Implementation Details

In Algorithm 1 we provide a structured overview of the meta learning algorithm for the e-OSVOS test time optimization. The optimization is defined by its trainable parameters $\boldsymbol{\theta}_g = \{\boldsymbol{\theta}_f^0, \boldsymbol{\lambda}\}$ consisting of the model initialization and neuron-level learning rates. Given a taskset $\mathcal{T}_{train}$ of training tasks we sample batches of tasks, fine-tune a model for $T$ iterations, and update $\boldsymbol{\theta}_g$ to achieve an optimal generalization to the respective test sets $\mathcal{D}_{test}^{n,k}$ of sequence $n$ and object $k$.

For the sake of completeness and in order to facilitate the reproduction of our results, we provide additional implementation details to Section 4.2 of the main paper. We do not apply regularization, such as weight decay or dropout, neither during the meta training nor at test time. To improve segmentation results, we double the default RoIAlign [1] pooling window size to 28. As the YouTube-VOS [3] validation dataset does not provide publicly available ground truth, we extract 100 sequences of the training set to monitor our meta learning progress.

---

**Algorithm 1:** Meta learning the e-OSVOS test time optimization

**Input:** Optimization $g$ with model initialization and neuron-level learning rates $\boldsymbol{\theta}_g = \{\boldsymbol{\theta}_f^0, \boldsymbol{\lambda}\}$. Meta learning rate $\beta$. Fine-tuning iterations $T$.
**Data:** Taskset $\mathcal{T}_{train}$
**Output:** $\boldsymbol{\theta}_g^*$

1   **while** *not done* **do**
2     Sample tasks $\mathcal{T}_{n,k} = \{\mathcal{D}_{train}^{n,k}, \mathcal{D}_{test}^{n,k}\} \sim \mathcal{T}_{train}$
3     **forall** $\mathcal{T}_{n,k}$ **do**
4       **for** $t \leftarrow 0$ **to** $T-1$ **do**
5         Update $\boldsymbol{\theta}_f^{t+1} = g(\nabla_{\boldsymbol{\theta}_f^t} \mathcal{L}_{seg}(\mathcal{D}_{train}^{n,k}, \boldsymbol{\theta}_f^t), \boldsymbol{\theta}_f^t)$
6       $\mathcal{L}_{BPTT}^{n,k} = \mathcal{L}_{seg}(\mathcal{D}_{test}^{n,k}, \boldsymbol{\theta}_f^T)$
7     Update $\boldsymbol{\theta}_g \leftarrow \boldsymbol{\theta}_g - \beta \nabla_{\theta_g} \sum_{\mathcal{T}_j} \mathcal{L}_{BPTT}^{n,k}$

---

Table 1: VOS performance comparison per sequence on the **DAVIS 2017 validation** benchmark between e-OSVOS and PReMVOS [2] which resembles the state-of-the-art for fine-tuning methods. We present the mean Intersection over Union $\mathcal{J}$ averaged over the number of objects indicated in parentheses.

| Method | Bike-Packing (2) | Blackswan | Bmx-Trees (2) | Breakdance | Camel | Car-Roundabout | Car-Shadow | Cows | Dance-Twirl | Dog | Dogs-Jump (3) | Drift-Chicane | Drift-Straight | Goat | Gold-Fish (5) | Horsejump-High (2) | India (3) | Judo (2) | Kite-Surf (3) | Lab-Coat (5) | Libby | Loading (3) | Mbike-Trick (2) | Motocross-Jump (2) | Paragliding-Launch (3) | Parkour | Pigs (3) | Scooter-Black (2) | Shooting (3) | Soapbox (3) |
|---|---|---|---|---|---|---|---|---|---|---|---|---|---|---|---|---|---|---|---|---|---|---|---|---|---|---|---|---|---|---|
| PReMVOS [2] | **77.1** | **95.8** | **60.9** | 80.8 | **92.6** | **97.5** | **97.1** | **94.1** | 79.8 | 94 | 88.1 | **88.2** | **94.9** | **88.8** | **85.6** | **83.9** | **54.5** | **83.1** | **44.5** | **60.4** | **86.8** | 78 | **80.7** | 53.2 | 30.4 | **92.4** | **69.6** | **82.6** | **77** | 75.6 |
| e-OSVOS-100-OnA | 73.4 | 93.1 | 55.2 | **82.1** | 89.7 | 96 | 96.1 | 92.4 | **86.5** | 94 | **89.2** | 87.9 | 93 | 88.1 | 85.5 | 81.3 | 54 | 81.2 | 35.9 | 59.8 | 84.8 | **78.2** | 79.3 | **69.7** | **44.8** | 91.7 | 68.3 | 80.7 | 72.1 | **83.1** |

| (a) Frame 1 (Ground Truth) | (b) Frame 2 | (c) Frame 40 |
|---|---|---|

Figure 1: Visualization of our e-OSVOS-50-OnA results on the *blackswan*, *kite-surf*, *shooting* and *india* sequences from the DAVIS 2017 validation set. We illustrate the output of the Mask R-CNN [1] mask head as well as the corresponding object detection bounding box predictions.

## 2 DAVIS 2017 Sequence Analysis

In Table 1, we provide a per sequence comparison between our e-OSVOS-100-OnA and PReMVOS [2] on the DAVIS 2017 validation set. The e-OSVOS-50-OnA method applies $T = 100$ first frame fine-tuning and subsequently $T_{OnA} = 10$ iterations on past mask predictions every fifth frame ($I_{OnA} = 5$). PReMVOS applies up to 1000 iterations and achieves state-of-the-art results with an ensemble of additional methods including a re-identification and contour snapping module. Nevertheless, we surpass the results of PReMVOS on many sequences while running orders of magnitude faster (0.29 vs. 0.01 frames per second). In Figure 1, we present visual results on four sequences for which we achieve worse results compared to PReMVOS. Due to the RoIAlign feature pooling in the Mask R-CNN [1] architecture and without an additional contour snapping, we fail to produce highly detailed masks as observable on the *blackswan* sequence (first row). Furthermore, the *kite-surf* (second row) and *india* (forth row) sequences demonstrate that our e-OSVOS approach is likely to yield false positive bounding box detections once an object is not visible anymore.