[Reviews · NeurIPS 2020]

Review 1

Summary and Contributions: The paper presents an approach for one-shot video object segmentation that replaces the pretraining step on large image segmentation datasets with a meta-task that learns an optimal network initialization for one-shot fine-tuning and other hyper-parameters such as learning rates. The claimed advantage is speed, evaluation on the DAVIS datasets demonstrates that the proposed approach has a better speed/accuracy tradeoff than other OSVOS approaches that employ finetuning but worse than the SOA network STM, that uses attention instead of finetuning.

Strengths: Novelty - to my understanding this is the first time that MAML inspired meta-learning is used for OSVOS. The results and the ablation study seems to demonstrate its effectiveness.

Weaknesses: While the paper’s major claim is efficiency, intended as inference time, there is no treatment of the topic neither in the paper nor in the supplemental material beside the plot of figure 1 (change power of ten to integer number in the x-axis notation). Why is the algorithm more efficient? Is the fine-tuning faster to converge because of the learned initialization? How many fine-tuning interactions are needed on pre-trained networks? How many with meta-learning? How accuracy changes increasing number of iterations with the proposed approach? Many of the methods that are compared in this paper strived for accuracy - I suspect their finetuning might have been much faster, reducing the number of iterations with negligible loss of accuracy. Also, is the runtime speed amortized over the number of frames? If so, the longer the video the less relevant is the time spent finetuning. Furthermore, an important baseline is missing. I would have liked to see the performance of the proposed approach, with all the bells and whistles and with pretraining on ImageNet or any other large scale image segmentation dataset instead of the proposed meta-learning. That’s crucial to assess the actual benefits of meta-learning compared to pretraining. Neuron-Level Learning Rates - to my understanding a learning rate is learned for each kernel and not for each neuron. If that’s the case, please improve the wording, which is confusing.

Correctness: The proposed method and evaluation is technically sound.

Clarity: The paper is well written.

Relation to Prior Work: The discussion of related works is comprehensive.

Reproducibility: Yes

Additional Feedback:


Review 2

Summary and Contributions: The paper proposes three improvements to finetuning-based video object segmentation. The contributions include 1) a meta learning strategy for learning a better initialization weight, 2) the learning rate is predicted with learnable parameters, and 3) the online finetuning is done on the detection model directly. *after rebuttal*, I have increased my rating to 6, as I believe it's overall an interesting combination of VOS and meta-learning. The rebuttal is also very helpful, especially fig1. i encourage the authors to include the missing details if accepted.

Strengths: The techniques proposed in the paper are quite intriguing. Especially, the idea of using meta-learning to learn a better initialization is well-motivated, and seems to be helpful as demonstrated in the experiments. The paper is well-presented, and the discussion with prior work is very comprehensive.

Weaknesses: The main concern is that, the validations in the paper are mostly done empirically, and lack of more in-depth analysis. Therefore, I am a bit skeptical of the effectiveness of the proposed methods, I believe more experimental analysis should be provided to confirm the efficacy. In particular, it would be interesting to demonstrate the difference of the weights learned with and without meta-learning. It might be also useful to show curves of J&F -- number of iterations in the test, to see how the meta-learning can help the finetuning in the test time. For the learning rate predictor, it would be interesting to see in which circumstances it predicts a larger learning rate, etc. Some qualitative results should be provided as videos, to clearly demonstrate the improvements provided by the proposed components. Currently, due to that the analysis is largely missing, it is difficult to gain further understanding of the proposed methods. Thus I believe the insights from this paper is a bit limited.

Correctness: Seems correct to me.

Clarity: The paper is well-written.

Relation to Prior Work: The contributions are clearly summarized.

Reproducibility: No

Additional Feedback:


Review 3

Summary and Contributions: The paper addresses the problem video object segmentation in the semi-supervised setting, meaning that the instance segmentation masks for the first frame are given during inference time. High performing methods in this field tend to fine-tune a model based on the first frame during inference, resulting in high latency. This paper proposes several tricks to reduce the fine-tuning overhead. One key idea is to meta learn the model initialization and fine-tuning learning rates for the test time optimization. In fact, every neuron in the network receives a separate learning rate. Moreover, the method fine-tunes on not only the first frame, but also subsequent frames using inferred masks. Experiments were done on DAVIS-2016/2017 + Youtube-VOS. These are all common benchmarks. The authors claim novelties in: - Learn the model initialization instead of just training an auxiliary model and start from a random checkpoint. - Learn the learning rates: a learning rate is learned for each neuron in the network. - It uses Mask R-CNN that first detect objects and infer the masks. This way, no global semantic segmentation is necessary. I agree with the authors' assessment regarding the contributions.

Strengths: - Problem statement: video object segmentation is a well researched problem in the computer vision community, and is therefore relevant to NeurIPS. - Novelty: AFAIK, I do not know any previous work that attempts to optimize for the model initialization and learning rates that are used to fine-tune for the first frame. I believe that this approach is indeed novel. The meta learning framework seems sound as well. - Sufficient experimentation: the paper demonstrated strong results on 3 benchmark datasets, and it has a good ablation in table 1 as well. - Strong results: the method outperforms other methods (with the exception of STM) by a large margin on all 3 datasets.

Weaknesses: There are some clarity related issues in the method section. The goal of the method is to learn the model initialization and the learning rates for the neurons. One typically cannot learn the learning rate. The reason why is because there is no trivial gradient flowing from the loss to the it. In this paper, the learning rates are learned. So why is there a gradient wrt the learning rate in the first place? I am positive that the authors have somehow solved the problem, but I have a hard time finding it in the paper. It's a shame, but without fully understanding how the gradients wrt to the meta parameters are derived, I cannot give a better score than borderline.

Correctness: Mostly correct, except for the clarity issue mentioned above.

Clarity: Some clarity issues as mentioned above. Otherwise, good presentation.

Relation to Prior Work: The paper gave a thorough outline of existing methods for video object detection, including methods with or without finetuning.

Reproducibility: Yes

Additional Feedback: ln 7: To mitigate what? The previous sentence says that most methods refrain from doing test time optimization. So mitigating refraining from optimization? Table 1 says fine-tune epochs? Since we are doing single-shot training, is one epoch equal to one image also? In ln 167, does g refer to the learning rate and number of steps? The benchmark datasets are all fairly small. Did the authors observe any overfitting when doing meta learning? **************************************************************************** POST REBUTTAL / REVIEWER DISCUSSION COMMENTS Thank you for providing the rebuttal. I am satisfied with the rebuttal, but I also concur with my fellow reviewers, R1 and R2, on the missing detailed latency analysis and ablation. I'll improve my rating slightly. Best of luck!


Review 4

Summary and Contributions: In this paper, an efficient approach of online learning based video object segmentation is proposed. The proposed method starts from OSVOS [6], and make some essential changes to make it efficient. The major changes are changing the backbone and adopting meta learning for initialization. The proposed method show competitive performance to the state-of-the-art methods on VOS benchmarks.

Strengths: The proposed method is based on online learning but efficient. Obviously, the recent trend in VOS is offline approaches that do not rely on online learning. it is because online learning has an important shortcoming that slows down the inference. In this paper, instead of following the recent trend that adopts the offline method, the authors approached to solve the disadvantages of the online method through meta learning. I want to admit that part. it is because the online learning based method is better than the offline learning method in some aspects. For example, it generalize to unseen objects better than offline methods as shown in Table 2, J unseen.

Weaknesses: The performance, in terms of both the accuracy and the speed, lags behind the recent state-of-the-art offline methods.

Correctness: The proposed methodology look correct.

Clarity: The paper is written clearly.

Relation to Prior Work: Yes.

Reproducibility: Yes

Additional Feedback: The implementation looks very challenging. The author is strongly encouraged to publish their source code.

[Author Response · NeurIPS 2020]

We thank all reviewers for their constructive and valuable feedback and are delighted to receive an overall positive
acceptance of our work. Furthermore, we will integrate all changes according to your suggestions and questions.

**Effectiveness (R1, R2)** In the context of our work, efficiency refers to the runtime of the test time optimiza-
tion which directly translates to inference time. In order to achieve high segmentation performance previous
fine-tuning approaches suffer from unfeasible high runtime due to many fine-tuning epochs (up to 1000). Our
approach reduces the number of epochs drastically by meta learning the initialization and learning rates. Fur-
thermore, in Figure 1 (which will be included in the supplementary) we illustrate the performance gains and in-
vertedly reduction in FPS of our e-OSVOS framework for increasing number of fine-tuning epochs. For many
fine-tuning epochs, the actual inference time on the frames is neglectable with respect to the fine-tuning. For
longer sequences the amortization of the fine-tuning time is higher, however, these sequences usually face addi-
11/12 tional challenges due to changing object appearance which can be tackled by the presented online adaptation.

**State-of-the-art (R1, R4)** We understand any potential
doubt of our overall impact on the field of VOS as we
merely provide a new state-of-the-art for fine-tuning ap-
proaches. However, considering publications of the recent
years, the VOS community has collectively deemed fine-
tuning as unfeasible. Therefore, we hope our demonstra-
tion of an efficient fine-tuning approach enabled through
meta learning will have a substantial impact and ignite
future research based on the release of our code base.

Figure 1: Evaluation on the DAVIS 2017 validation set.

**Learning (neuron-level) learning rates (R1)** We pre-
dict test time learning rates for fully-connected and convo-
lutional layers, each consisting of neurons with a weight
vector and a scalar bias. To account for convolutional
neurons with a spatial weight tensor (also referred to as
kernels), we will rename *weight vector* to *weight tensor*.

**(R3)** Learning and deriving a gradient with respect to the learning rates $\boldsymbol{\lambda}$ is analogous to the model initialization $\boldsymbol{\theta}_f^0$.
Hence, we formulate the meta optimization of the optimization $g$ for the joint set of its parameters $\boldsymbol{\theta}_g = \{\boldsymbol{\theta}_f^0, \boldsymbol{\lambda}\}$. The
gradient flows from the loss (Equation (4)) to each of the $T$ parameter updates (Equation (5)) of the inner fine-tuning
loop. The connection between the inner and outer optimization is also illustrated in Algorithm 1 of the supplementary.
As the SGD update consists only of differentiable operations, gradients with respect to the learning rates can be derived
analogous to the derivation for the initial parameters. It should be noted, that these gradients are with respect to a
different loss ($\mathcal{L}_{seg}(\mathcal{D}_{test}, \boldsymbol{\theta}_f^T)$) as the ones of the inner gradient ($\mathcal{L}_{seg}(\mathcal{D}_{train}, \boldsymbol{\theta}_f^t)$). It is a common approach to meta
learn the initialization and learning rate(s) jointly. We refer the reader to [2, 1] for further insights. **(R2)** At test time,
the set of learning rates does not change and is the same for all sequences. However, it is interesting to observe which
neurons are updated with particularly small, e.g., biases of last layers, or large, e.g., FC6 of the box head, learning
rates. The FC6 layer of the box head prepares the spatial bounding box features for the regression and classification
heads and benefits from a strong adaption to each individual given object. To further illustrate e-OSVOS, we will add a
summarized visualization of the overall more than 20000 neuron-level learning rates to the supplementary.

**Other comments/suggestions (R1)** In the ablation study (first row of Table 1), we present a Mask R-CNN baseline
without meta learning which was pre-trained on ImageNet, COCO segmentation, YouTube-VOS and DAVIS 2017.
However, this baseline is not representative for state-of-the-art fine-tuning approaches as we omitted any additional
handcrafted test time improvements. For a state-of-the-art fine-tuning approach without meta learning but with bells
and whistles, e.g., online adaption, we compare to OnAVOS [3] in Table 3. **(R2)** Upon acceptance we will publish
our results on the official DAVIS challenge webpage which provides a tool for visual comparison per sequence, e.g.,
blackswan. **(R3)** The *mitigate* in line 7 refers to *shortcomings* and we will improve the understandability of the abstract.
The fine-tuning epochs in Table 1 refer to a single update with one image. However, to improve the generalization over
a sequence, we train on batches of random transformations of that image. The superscript of $\boldsymbol{\theta}_f$ always implies how
the parameters were optimized, e.g., for $T$ update steps or by optimization $g$. In our formulation, the optimization $g$
describes a model initialization, a set of learning rates and number of steps $T$. We will clarify this overloading of the
superscript in the final version. Furthermore, we observed overfitting without the YouTube-VOS dataset and therefore
train on a combination of all three datasets.

[1] Antreas Antoniou, Harrison Edwards, and Amos Storkey. How to train your MAML. In *International Conference on Learning Representations*, 2019.

55/56 [2] Chelsea Finn, Pieter Abbeel, and Sergey Levine. Model-agnostic meta-learning for fast adaptation of deep networks. In *International Conf. on Machine Learning*, ICML'17, pages 1126–1135. JMLR.org, 2017.

[3] Paul Voigtlaender and Bastian Leibe. Online adaptation of convolutional neural networks for video object segmentation. In *BMVC*, 2017.


[Meta-Review · NeurIPS 2020]

All four reviewers feel borderline about the paper. Everyone was positive about the idea of combining MAML-inspired meta-learning for one-shot video object segmentation. Reviewers shared concerns that they would have like to see more analysis by the paper to offer more insight and understanding to the gains offered by the proposed method. The authors are recommended to take advantage of the feedback from the reviewers to revise their paper for the camera-ready.